# Polynomial Input Preconditioning for Zero-Shot Time Series Forecasting

**Jerry Han** [1]  **Alex Kawaja** [1]  **Benjamin Cole** [1]  **Elad Hazan** [1]

## Abstract

Universal Sequence Preconditioning (USP) is a recent theoretical framework for online sequence prediction that preprocesses observations via a fixed polynomial convolution, yielding hidden-dimension-free sublinear regret for marginally stable linear dynamical systems. This paper investigates whether these techniques transfer empirically to zero-shot patch-based transformer time-series forecasters such as Moirai 2.0. We propose **polynomial input preconditioning**, where we concatenate the preconditioned signal as an auxiliary input channel while leaving the forecast target unchanged. By only adding $0.11\%$ parameters, our method improves a Moirai 2.0 Small baseline by $2.9\%$ geometric-mean MASE on GIFT-Eval and achieves similar gains on FEV-Bench. At a 100K-step budget, the method remains substantially better than the matched baseline, with a larger relative gap and lower cross-seed variance. Polynomial input preconditioning performs better with longer prediction horizon lengths, attaining a $5.4\%$ improvement on long horizon prediction tasks. Capacity-matched zero and duplicate-channel controls show that the gain comes from the polynomial content rather than extra parameters. These results suggest that time-series foundation models can benefit substantially from theory-guided structure injected before tokenization, offering a lightweight alternative to improving model performance without scaling model size or attention complexity.

## 1. Introduction

**Universal sequence preconditioning.**  In 2025, based on the theory of linear dynamical systems, Marsden & Hazan (2025) proposed Universal Sequence Preconditioning (USP) for online learning. The technique generalizes differencing; in particular, they suggest preprocessing the inputs via a fixed convolution of polynomial coefficients,

$$x_t^{\mathrm{pre}} \;=\; \sum_{k=0}^{d} c_k\, x_{t-k},$$

where $c_k$ are fixed coefficients and $d$ is the polynomial degree. They prove that this can result in hidden-dimension-free (up to logarithmic factors) sublinear regret with respect to the best LDS predictor.

**Does this transfer to zero-shot time-series forecasting?** We investigate how the USP framework can be applied to pretrained time series foundation models, and specifically to the Moirai 2.0 model. The setting of pretrained time series foundation models (Woo et al., 2024; Liu et al., 2025; Das et al., 2024; Ansari et al., 2024; Goswami et al., 2024; Auer et al., 2025) differs from USP in two important ways. First, the forecaster is trained offline on a large corpus and deployed for zero-shot prediction over a multi-step horizon, rather than making one-step predictions in an online loop where the true observation is revealed after each step. Second, the training data comes from many heterogeneous sources that may not be well-modelled as finite-dimensional LDSs.

**Our approach.**  A direct way to use USP would be to train the forecaster to predict the preconditioned target sequence and then invert the preconditioning filter to recover forecasts in the original scale. For multi-step forecasting, this inversion may lead to error accumulation as each recovered future value depends on earlier recovered values, so forecast errors are fed back into later predictions. To address this we can leave the forecast target unchanged and concatenate the preconditioned signal as an auxiliary input channel alongside the raw target and the observation mask. Because the model never operates in preconditioned space at inference, no inversion is needed and error accumulation is avoided.

A second motivation comes from the architecture of Moirai 2.0 itself. Patch-based transformers (Nie et al., 2023; Dosovitskiy et al., 2021) split the input into non-overlapping length-$P$ windows and embed each patch independently,

---
[1]Department of Computer Science, Princeton University, Princeton, NJ, USA. Correspondence to: Jerry Han <jh1161@princeton.edu>.

*Proceedings of the $2^{nd}$ ICML Workshop on Foundation Models for Structured Data*, Seoul, South Korea. 2026. Copyright 2026 by the author(s).

creating a cross-patch information bottleneck: before the first self-attention layer, each patch embedding encodes only $P$ time steps, so multi-patch correlations must be discovered entirely through attention. A polynomial preconditioner that mixes information across the same multi-patch scales before patching can inject cross-patch structure into every embedding at negligible parameter cost.

## 2. Method

**Preconditioning polynomial.** Given a normalized input series $x$, let $\tilde{T}_d(z) = z^d + c_1 z^{d-1} + \cdots + c_d$ be the degree-$d$ monic Chebyshev polynomial in descending powers. Following the implementation convention, the leading monic term is the current value and the remaining coefficients are applied at stride-$s$ lags:

$$r_t = x_t^{\text{pre}} - x_t = \sum_{k=1}^{d} c_k \, x_{t-ks}. \tag{1}$$

where $s$ is a stride parameter set equal to the patch size $P$. Our default is the monic Chebyshev polynomial of the first kind, $\tilde{T}_d$, with $d = 4$. For $P = 16$ the coefficients are $(c_1, \ldots, c_4) = (0, -1, 0, \frac{1}{8})$, so $r_t = -x_{t-32} + \frac{1}{8} x_{t-64}$.

**Preconditioning channel.** The residual $r$ is partitioned into patches $\mathbf{r}_i$ aligned with the data patches $\mathbf{p}_i$. In Moirai 2.0, each patch passes through a per-patch input projection (a two-layer residual MLP with SiLU activation) that maps the concatenation of the data patch and an observation mask to an embedding vector. We widen this projection to accept a third input, the preconditioning patch, and so the input dimension grows from $2P$ to $3P$, adding 12K parameters (0.11% of Moirai 2.0 Small). The transformer decoder is unchanged. We refer to this default ($d = 4$ Chebyshev, no dropout) as d4, and to the same architecture trained with 10% per-patch channel dropout as d4_dropout. The $z$-score normalization that precedes the residual computation uses only observed context values; prediction-window positions are excluded, so no future information leaks into $r_t$.

## 3. Experiments

**Setup** We train Moirai 2.0 Small (Liu et al., 2025) (11.4M params, 6-layer transformer, $P{=}16$) from scratch on the public LOTSA corpus (Woo et al., 2024) for 10K steps (batch 256, AdamW, cosine LR $10^{-3}$, 1K warmup, bf16). The official Moirai 2.0 model was trained for 100K steps on 295B observations, while our accessible LOTSA setup contains 27B observations and is run for either 10K or 100K steps.

We then train models d4 and d4_dropout with the architectural modifications described above. For d4_dropout,

the preconditioning channel is independently zeroed with probability 0.1 per patch during training and always provided at inference. All trained models share identical data, optimiser, seeds, and compute, so the effects we report are controlled relative improvements.

**Evaluation** At inference, the model predicts multiple future patches per step. For longer horizons, we proceed with autoregressive decoding, where the median forecast patch is appended to the context and the decoder is re-run with the preconditioning residual recomputed over the extended context. MASE is also computed from the median forecast.

Evaluation uses GIFT-Eval (Aksu et al., 2024) (97 dataset×horizon configs) at context length 4,000. GIFT-Eval is a large-scale zero-shot benchmark for general and foundation time-series forecasters, spanning 23 datasets, over 144,000 time series, seven domains, ten sampling frequencies, multivariate inputs, and short- to long-horizon tasks. We report normalized MASE following the GIFT-Eval leaderboard convention: each configuration's MASE is divided by the seasonal naive baseline's MASE, then aggregated via geometric mean. We run 5 seeds (0, 1, 2, 7, 42) and report cross-seed mean and standard deviation.

We also evaluate with the independent FEV-Bench (Shchur et al., 2025) (100 tasks), which is a realistic forecasting benchmark for reproducible and statistically sound model comparison, comprising 100 tasks from 96 datasets across seven real-world domains, including covariate-rich univariate and multivariate settings.

**Conditions.** We compare four conditions, all sharing the same training data, optimizer, and seeds. **Baseline** is unmodified Moirai 2.0 (no preconditioning channel). **d4** and **d4_dropout** add the polynomial preconditioning channel of §2 (without and with 10% per-patch channel dropout). **Zero** and **Duplicate** are capacity-matched controls: they share d4_dropout's widened input projection and 12K extra parameters, but feed the third channel either all zeros (Zero) or a copy of the data channel (Duplicate). This helps us determine whether changes in performance are due to the polynomial content or just from the extra capacity.

### 3.1. Main Results

Both d4 and d4_dropout improve over Baseline on every seed (Table 1). The average improvements across seeds are 2.8% and 2.9% respectively. Their average MASE values are within 0.1% of each other, so the bulk of the gain comes from the polynomial content rather than the dropout. Zero and Duplicate share d4_dropout's architecture and 12K-parameter overhead. Duplicate hurts (39/97 wins) and Zero has no significant effect (54/97, $p = 0.31$). This demonstrates the importance of the polynomial content of the filter

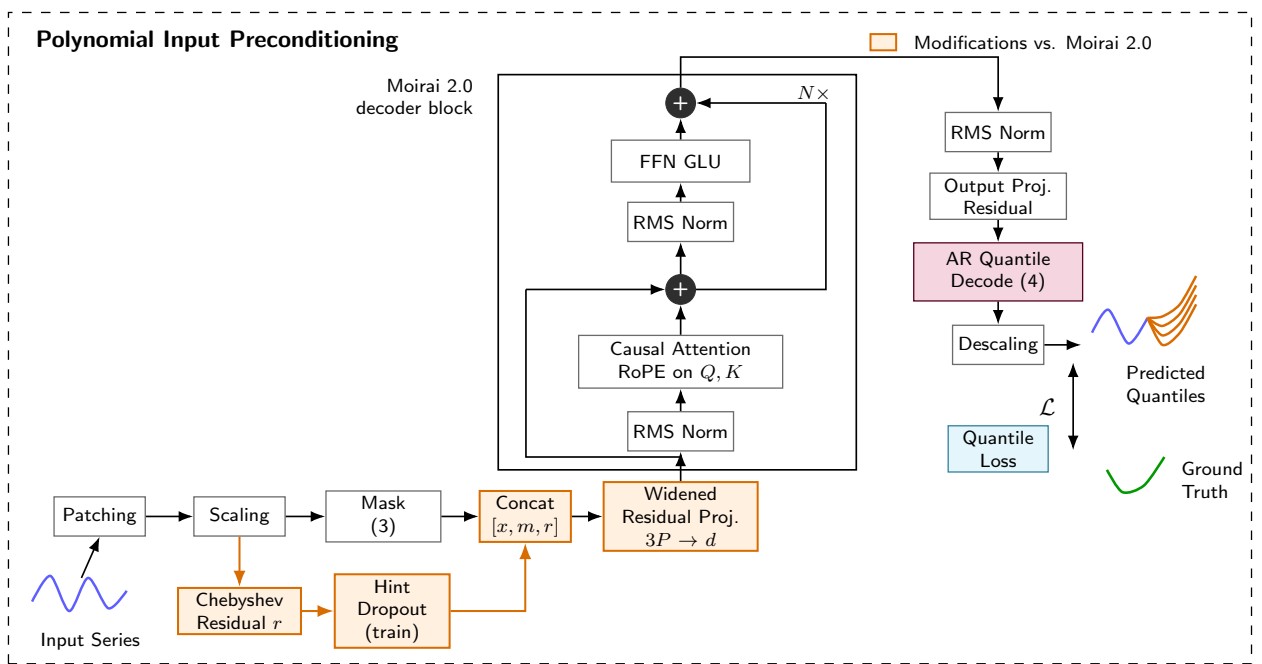

*Figure 1.* Polynomial input preconditioning in Moirai 2.0. Orange boxes mark components changed relative to the standard architecture: the Chebyshev residual channel, optional training-time hint dropout, concatenation of $[x, m, r]$, and the widened residual input projection from $2P$ to $3P$ dimensions. The decoder block and autoregressive quantile output head are unchanged. Total overhead: 12K parameters (0.11%).

*Table 1.* Main results on GIFT-Eval (5 seeds). MASE is averaged per configuration across seeds, then aggregated via geometric mean. Wins: configurations where the method's mean MASE < Baseline's (out of 97); $p$-value is calculated from a two-sided paired sign test.

| Method | MASE | $\Delta$% | Wins | $p$ |
|---|---|---|---|---|
| **d4_dropout (Ours)** | **0.837** | $-2.9$% | **72/97** | $< 10^{-5}$ |
| d4 (Ours) | 0.838 | $-2.8$% | 72/97 | $< 10^{-5}$ |
| Zero ctrl | 0.858 | $-0.5$% | 54/97 | 0.31 |
| Baseline | 0.862 | — | — | — |
| Duplicate ctrl | 0.865 | $+0.4$% | 39/97 | 0.07 |

*Table 2.* FEV-Bench (5 seeds each). SQL (scaled quantile loss) is FEV-Bench's primary metric. MASE values are unnormalized (unlike the normalized MASE in Table 1). Wins: per-task mean MASE of d4_dropout < Baseline (out of 100); $p$ from paired sign test.

| Method | MASE | SQL | Wins | $p$ |
|---|---|---|---|---|
| **d4_dropout (Ours)** | **1.254** | **1.024** | **78/100** | $< 10^{-7}$ |
| Baseline | 1.282 | 1.045 | — | — |
| $\Delta$% | $-2.2$% | $-1.9$% | | |

*Table 3.* Fixed vs learned coefficients (5 seeds, 10K steps).

| Setting | MASE | Std | $\Delta$% |
|---|---|---|---|
| **Fixed Chebyshev** | **0.837** | 0.008 | $-2.9$% |
| Learned, zero init | 0.839 | 0.006 | $-2.6$% |
| Learned, Cheb. init | 0.844 | 0.007 | $-2.1$% |

rather than just the additional capacity.

On FEV-Bench (Table 2, 5 seeds each), d4_dropout wins on 78/100 tasks with $\Delta = -2.2$% MASE ($p < 10^{-7}$), confirming the benefit is not benchmark-specific.

### 3.2. Ablations

We perform a sweep over the Chebyshev polynomial degree $d$ from $d = 2$ to $d = 7$, over 5 seeds each (Figure 2). Every tested degree improves over the baseline, and $d = 4$ gives the lowest mean MASE. The performance does not depend too strongly on the degree, so we don't have to sharply tune the degree as a hyperparameter.

We also consider directly learning the preconditioning coefficients (Table 3). We perform two different initializations:

one from the Chebyshev coefficients and one from zero. Both learned variants beat Baseline, but fixed Chebyshev still performs best, suggesting it is a good prior.

We also compare Chebyshev coefficients with coefficients obtained from Legendre polynomials, EMA, and finite differences and find that Chebyshev performs the best (Appendix D). We also perform other stride and dropout ablations.

**Horizon stratification.** On short horizons (Prediction Length (PL) $\leq 60$) d4 and d4_dropout are within 0.4%

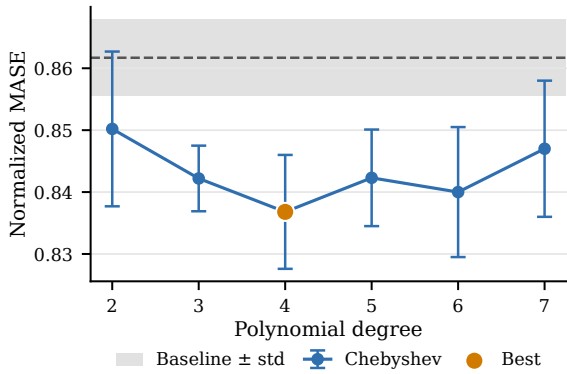

*Figure 2.* Chebyshev polynomial degree sweep (10K steps, stride $s=16$, 10% channel dropout). Filled blue circles are 5-seed means with $\pm 1$ std error bars. The grey band is the baseline 5-seed mean $\pm 1$ std.

*Table 4.* Horizon stratification (10K steps, 5 seeds). Geomean MASE; $\Delta\%$ relative to BL within each bin.

| Term | $n$ | BL | d4 ($\Delta$) | d4_dropout ($\Delta$) |
|------|-----|------|----------------|------------------------|
| short | 55 | 0.840 | **0.825** (−1.8%) | 0.828 (−1.4%) |
| medium | 21 | 0.859 | 0.830 (−3.4%) | **0.823** (−4.2%) |
| long | 21 | 0.924 | 0.880 (−4.8%) | **0.874** (−5.4%) |

of each other (Table 4). On medium and long horizons (PL $\geq$ 480) d4_dropout beats d4 by 0.7–0.85% relative MASE. The performance increase over baseline is greatest for the long horizon prediction tasks, where we attain a 5.4% MASE reduction. This pattern is consistent with our hypothesis that dropout makes d4_dropout more robust when the preconditioning signal becomes partially synthetic at longer horizons.

### 3.3. Extended Training

At 100K optimizer steps (Table 5, 5 seeds) the relative MASE reduction increases to −3.9% while both capacity controls now do worse than baseline (Duplicate +0.9%, Zero +1.7%). d4_dropout also shows lower cross-seed variance: raw standard deviation is 0.0079 versus 0.0189 for Baseline, equivalent to 0.9% versus 2.2% of mean MASE.

**Training dynamics.** The training-loss curves in Fig. 3 separate early and remain ordered throughout the run, with the gap widening after roughly 20K steps. This is consistent with the 100K evaluation results too. The preconditioning channel is not merely a short-horizon optimization aid, but continues to improve the learned predictor as training proceeds.

**Different warmup schedules.** We also train with 10K warmup (the official Moirai 2.0 schedule) instead of 1K, across 5 seeds and 100K total steps. d4_dropout improves by −3.0% over Baseline under this schedule (Ta-

*Table 5.* 100K training budget, 5 seeds. MASE is averaged per configuration across seeds, then aggregated via geometric mean.

| Method | MASE | $\Delta\%$ | Std |
|--------|------|------------|-----|
| **d4_dropout (Ours)** | **0.844** | −3.9% | 0.008 |
| Baseline | 0.878 | — | 0.019 |
| Duplicate ctrl | 0.886 | +0.9% | 0.008 |
| Zero ctrl | 0.894 | +1.7% | 0.035 |

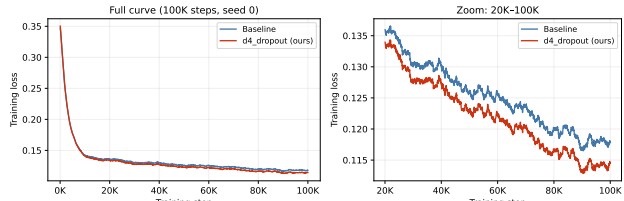

*Figure 3.* Training loss on LOTSA (seed 0, 100K steps, EMA-smoothed). Left: full curve. Right: zoom on steps 20K–100K. d4_dropout consistently tracks below Baseline throughout training, with the gap widening in the later stages.

ble 15 in Appendix E), consistent with the −3.9% observed under our default schedule.

## 4. Discussion and Conclusion

In this paper, we adapt the ideas presented in Marsden & Hazan (2025) to zero-shot patch-based time-series transformers. Concatenating the preconditioned signal as an auxiliary channel avoids the error accumulation of target-side preconditioning at multi-step horizons, and adds only 0.11% parameters, while providing a significant 2.9% improvement on GIFT-Eval. At a 100K-step budget, the method remains substantially better than the matched baseline, with a larger relative gap and lower cross-seed variance. The improvement is also largest at longer horizon lengths.

More broadly, these results suggest that time-series foundation models can still benefit substantially from simple, theory-guided input transformations. A fixed polynomial channel derived from USP theory improves zero-shot forecasting across benchmarks without significantly increasing parameter count or changing the transformer stack. This points to a practical path for making compact forecasters more accurate and parameter-efficient by injecting structure before tokenization, rather than relying only on larger models or heavier attention mechanisms.

The same principle may extend beyond time series. Pre-trained language models and other sequence models could benefit from lightweight, fixed preconditioning channels that expose long-range structure before tokenization or early embedding layers, offering a complementary alternative to scaling model size alone.

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

# A. Detailed Methodology

## A.1. Training Protocol

All experiments use identical training infrastructure on NVIDIA H100/A100 GPUs. The complete configuration:

| Parameter | Value |
|---|---|
| Architecture | Moirai 2.0 Small (6 layers, $d = 384$, $d_{\text{ff}} = 1024$, $P = 16$) |
| Total parameters | 11.4M (baseline), 11.4M + 12K (`d4_dropout`) |
| Training data | Public LOTSA setup, 27B observations |
| Official reference | 100K steps, 295B observations |
| Steps (10K runs) | 100 epochs $\times$ 100 batches = 10,000 steps |
| Steps (100K runs) | 1,000 epochs $\times$ 100 batches = 100,000 steps |
| Batch size | 256 |
| Optimizer | AdamW ($\beta_1 = 0.9$, $\beta_2 = 0.98$, $\epsilon = 10^{-6}$, wd $= 0.1$) |
| LR schedule | Cosine annealing, peak $\eta = 10^{-3}$, min $\eta = 0$ |
| Warmup | 1,000 steps (linear) |
| Precision | bf16 mixed precision |
| Anomaly filtering | $z$-score threshold = 8.0, variance ratio = 0.0 (disabled) |
| Seeds | 0, 1, 2, 7, 42 (5 seeds for main results) |

## A.2. Evaluation Protocol

All evaluations use the GIFT-Eval benchmark (Aksu et al., 2024) with 97 dataset$\times$horizon configurations:

- Context length: 4,000 tokens

- Patch size: $P = 16$

- Batch size: 64

- Metric: Normalized MASE (each configuration's MASE divided by the seasonal naive baseline's MASE)

- Aggregation: geometric mean across 97 configurations

- Reported quantities: cross-seed mean and standard deviation, per-seed MASE, and per-configuration win counts (out of 97 per seed; $5 \times 97 = 485$ when pooled across seeds)

## A.3. Preconditioning Channel Implementation

The monic Chebyshev polynomial $\tilde{T}_d(z) = z^d + c_1 z^{d-1} + \cdots + c_d$ is expanded in the monomial basis to obtain the coefficients used by the preconditioning filter. The preconditioning residual for time step $t$ is:

$$r_t = \left( x_t + \sum_{k=1}^{d} c_k\, x_{t-ks} \right) - x_t$$

This is computed over the full (normalized, zero-mean unit-variance) time series before patching. Unobserved time steps are masked to zero. During training with channel dropout $p$, each patch's preconditioning channel independently has probability $p$ of being replaced with zeros. At inference, the full preconditioning channel is always provided.

The preconditioning channel is concatenated with the target and mask channels before the input projection:

$$\mathbf{e}_i = \text{ResBlock}_{3P \to d}([\mathbf{p}_i; \mathbf{m}_i; \mathbf{r}_i])$$

The input projection ResBlock consists of:

$$\mathbf{z} = \text{SiLU}(W_1 \mathbf{x} + b_1) \qquad\qquad W_1 \in \mathbb{R}^{d \times 3P} \qquad\qquad (2)$$

$$\mathbf{o} = W_2 \mathbf{z} + b_2 + W_3 \mathbf{x} + b_3 \qquad\qquad W_2 \in \mathbb{R}^{d \times d},\ W_3 \in \mathbb{R}^{d \times 3P} \qquad\qquad (3)$$

Only $W_1$ and $W_3$ are wider than the baseline. $W_2$ and all downstream layers are identical.

**TransformerDecoder.** Moirai 2.0 Small uses a 6-layer decoder-only transformer with $d_{\text{model}} = 384$, $d_{\text{ff}} = 1024$, 6 attention heads, causal multi-head self-attention with rotary position embeddings (RoPE), SwiGLU feed-forward blocks, and RMSNorm. The output projection is another ResBlock mapping $\mathbb{R}^d \to \mathbb{R}^{K \times |\mathcal{Q}| \times P}$ where $K{=}4$ is the multi-token horizon and $|\mathcal{Q}|{=}9$ the number of quantile levels.

## B. Per-Seed Results

*Table 6.* d4 and d4_dropout vs. Baseline: per-seed pairwise comparison (10K steps, $\eta = 10^{-3}$).

| Seed | BL | d4 | d4_dropout | d4 $\Delta\%$ | d4_drop $\Delta\%$ |
|------|-----|-----|------------|---------------|--------------------|
| 0 | 0.8666 | 0.8327 | 0.8234 | $-3.91\%$ | $-4.99\%$ |
| 1 | 0.8503 | 0.8435 | 0.8465 | $-0.80\%$ | $-0.44\%$ |
| 2 | 0.8630 | 0.8283 | 0.8391 | $-4.02\%$ | $-2.77\%$ |
| 7 | 0.8676 | 0.8422 | 0.8320 | $-2.93\%$ | $-4.11\%$ |
| 42 | 0.8614 | 0.8425 | 0.8430 | $-2.19\%$ | $-2.13\%$ |
| **Mean** | **0.8617** | **0.8379** | **0.8368** | $-2.77\%$ | $-2.89\%$ |

Per-seed MASE for all conditions:

*Table 7.* Per-seed results. Both d4 and d4_dropout improve over Baseline and controls on every seed.

| Seed | Baseline | d4 (Ours) | d4_dropout (Ours) | Zero | Duplicate |
|------|----------|-----------|-------------------|------|-----------|
| 0 | 0.8666 | 0.8327 | **0.8234** | 0.8586 | 0.8777 |
| 1 | 0.8503 | **0.8435** | 0.8465 | 0.8738 | 0.8546 |
| 2 | 0.8630 | **0.8283** | 0.8391 | 0.8425 | 0.8662 |
| 7 | 0.8676 | 0.8422 | **0.8320** | 0.8573 | 0.8602 |
| 42 | 0.8614 | 0.8425 | **0.8430** | 0.8569 | 0.8661 |
| **Mean** | 0.8617 | 0.8379 | **0.8368** | 0.8578 | 0.8650 |

## C. Training Loss Analysis

*Table 8.* Final training loss across 5 seeds at 10K steps ($\eta = 10^{-3}$). Training loss differences are small ($\sim$1%), while eval MASE differences are much larger ($\sim$3%).

| Method | s0 | s1 | s2 | s7 | s42 | Mean | Std | Eval MASE |
|--------|-----|-----|-----|-----|------|------|-----|-----------|
| Baseline | .1366 | .1365 | .1358 | .1366 | .1365 | .1364 | .0003 | 0.8617 |
| d4 (ours) | .1356 | .1348 | .1354 | .1353 | .1341 | .1350 | .0006 | 0.8379 |
| **d4_dropout (ours)** | .1354 | .1350 | .1350 | .1353 | .1343 | **.1350** | .0004 | **0.8368** |
| Zero | .1378 | .1369 | .1363 | .1366 | .1353 | .1366 | .0008 | 0.8578 |
| Duplicate | .1376 | .1366 | .1370 | .1365 | .1355 | .1366 | .0007 | 0.8650 |
| **d4_dropout $\Delta\%$ vs BL** | | | | | | $-1.0\%$ | | $-2.9\%$ |

*Table 9.* Final training loss across seeds at 100K steps ($\eta = 10^{-3}$). d4_dropout maintains lower training loss on every seed.

| Method | s0 | s1 | s2 | s7 | s42 | Mean | Std | Eval MASE |
|--------|-----|-----|-----|-----|------|------|-----|-----------|
| Baseline | .1185 | .1185 | .1190 | .1194 | .1179 | .1187 | .0005 | 0.8784 |
| **d4_dropout (ours)** | .1144 | .1170 | .1155 | .1172 | .1155 | **.1159** | .0010 | **0.8443** |
| Zero | .1186 | .1188 | .1184 | .1188 | .1187 | .1187 | .0002 | 0.8937 |
| Duplicate | .1192 | .1193 | .1185 | .1197 | .1182 | .1190 | .0006 | 0.8860 |
| **d4_dropout $\Delta\%$ vs BL** | | | | | | $-2.4\%$ | | $-3.9\%$ |

# D. Ablations

All ablations use seed 0 with 10% channel dropout unless otherwise noted; the polynomial-degree sweep is the exception and is reported over 5 seeds each.

### D.1. Polynomial Degree (Chebyshev, $s = 16$, 10% dropout)

Residual lags count the nonzero entries of the actual hint coefficients $(c_1, \ldots, c_d)$ after the identity term has been stripped. With stride $s = P$, a nonzero $c_k$ contributes one residual lag at $kP$; zero coefficients do not count.

*Table 10.* Polynomial degree sweep (5 seeds, 10K steps, Chebyshev, $s = 16$, 10% dropout).

| Degree | Residual lags | MASE (mean $\pm$ std, $n$=5) | $\Delta$% vs BL |
|---|---|---|---|
| 2 | 1 (at $2P$) | $0.8502 \pm 0.0125$ | $-1.33\%$ |
| 3 | 1 (at $2P$) | $0.8422 \pm 0.0053$ | $-2.27\%$ |
| **4** | 2 (at $2P, 4P$) | $\mathbf{0.8368 \pm 0.0092}$ | $\mathbf{-2.89\%}$ |
| 5 | 2 (at $2P, 4P$) | $0.8423 \pm 0.0078$ | $-2.25\%$ |
| 6 | 3 (at $2P, 4P, 6P$) | $0.8400 \pm 0.0105$ | $-2.52\%$ |
| 7 | 3 (at $2P, 4P, 6P$) | $0.8470 \pm 0.0110$ | $-1.71\%$ |
| Baseline | — | $0.8617 \pm 0.0062$ | — |

Every tested degree beats the baseline, with $d = 4$ giving the lowest mean MASE. The curve is shallow, so we treat $d = 4$ as a strong default rather than as evidence of a sharply tuned optimum.

### D.2. Stride (Chebyshev $d = 4$, 10% dropout)

*Table 11.* Stride ablation (seed 0, 10K steps, Chebyshev $d = 4$, 10% dropout).

| Stride | MASE | $\Delta$% vs BL | Wins/97 |
|---|---|---|---|
| **16 ($= P$)** | **0.8234** | $-4.99\%$ | **66** |
| 4 | 0.8406 | $-3.00\%$ | 57 |
| 8 | 0.8518 | $-1.71\%$ | 62 |
| 32 | 0.8484 | $-2.10\%$ | 63 |

Stride $s = 16$ (equal to patch size $P$) is optimal: the nonzero coefficients align exactly with patch boundaries, providing maximally informative cross-patch lookback.

### D.3. Basis ($d = 4$, $s = 16$, 10% dropout)

*Table 12.* Basis ablation (seed 0, 10K steps, $d = 4$, $s = 16$, 10% dropout).

| Basis | MASE | $\Delta$% | Wins vs BL | Wins vs Cheby |
|---|---|---|---|---|
| **Chebyshev** | **0.8234** | $-4.99\%$ | 66/97 | — |
| EMA | 0.8329 | $-3.89\%$ | 65/97 | 41/97 |
| Legendre $d = 4$ | 0.8400 | $-3.07\%$ | 62/97 | 30/97 |
| Legendre $d = 6$ | 0.8397 | $-3.10\%$ | 64/97 | 50/97 |
| Finite Diff | 0.8569 | $-1.12\%$ | 54/97 | 29/97 |
| Duplicate | 0.8583 | $-0.96\%$ | 42/97 | 27/97 |

Chebyshev is best in this seed-0 sweep, while EMA is also competitive and finite differences are weakest.

### D.4. Learned Coefficients (5 seeds)

*Table 13.* Learned coefficient ablation (5 seeds, 10K steps). Per-seed learned coefficients and MASE.

| Seed | Learned $(c_1, c_2, c_3, c_4)$ | MASE | $\Delta\%$ |
|------|-------------------------------|------|------------|
| **Chebyshev init** (mean MASE = 0.844) | | | |
| 0 | $[-0.026, -1.032, -0.123, +0.007]$ | 0.833 | $-3.9\%$ |
| 1 | $[+0.010, -1.034, -0.134, -0.000]$ | 0.843 | $-0.8\%$ |
| 2 | $[-0.016, -1.027, -0.152, -0.011]$ | 0.856 | $-4.0\%$ |
| 7 | $[+0.002, -1.028, -0.141, +0.020]$ | 0.842 | $-2.9\%$ |
| 42 | $[-0.032, -1.030, -0.148, -0.015]$ | 0.846 | $-2.2\%$ |
| **Zero init** (mean MASE = 0.839) | | | |
| 0 | $[+0.308, +0.177, +0.057, +0.045]$ | 0.836 | $-3.6\%$ |
| 1 | $[-0.286, -0.188, -0.054, -0.044]$ | 0.847 | $-0.4\%$ |
| 2 | $[-0.323, -0.190, -0.058, -0.066]$ | 0.830 | $-4.0\%$ |
| 7 | $[-0.303, -0.168, -0.057, -0.048]$ | 0.839 | $-3.3\%$ |
| 42 | $[-0.331, -0.163, -0.028, -0.057]$ | 0.842 | $-2.2\%$ |
| **Fixed Chebyshev** (mean MASE = 0.837, not learned) | | | |
| 0 | $[0, -1, 0, +0.125]$ | 0.823 | $-5.0\%$ |
| 1 | $[0, -1, 0, +0.125]$ | 0.847 | $-0.4\%$ |
| 2 | $[0, -1, 0, +0.125]$ | 0.839 | $-2.8\%$ |
| 7 | $[0, -1, 0, +0.125]$ | 0.832 | $-4.1\%$ |
| 42 | $[0, -1, 0, +0.125]$ | 0.843 | $-2.1\%$ |

All three settings beat Baseline, but fixed Chebyshev (0.837) is best. The Chebyshev-init solution consistently keeps $c_2 \approx -1.03$ (the dominant 2-patch lookback) but drifts at other lags, particularly adding $c_3 \approx -0.14$ (a 3-patch term absent from the fixed polynomial). The zero-init solution converges to a smoothly decaying pattern whose sign is seed-dependent (4/5 seeds find negative coefficients, 1/5 finds positive) but whose magnitude is consistent ($|c_1| \approx 0.31$, $|c_2| \approx 0.18$). Despite having strictly more capacity, neither learned variant matches fixed Chebyshev, confirming that the Chebyshev structure provides regularization value.

## E. 100K Extended Training

*Table 14.* Full 100K training budget with capacity controls (5 seeds). `d4_dropout` vs BL: $-3.9\%$; `d4_dropout` vs Dup: $-4.7\%$; `d4_dropout` vs Zero: $-5.5\%$.

| | s0 | s1 | s2 | s7 | s42 | Mean |
|---|------|------|------|------|------|------|
| Baseline | 0.9092 | 0.8920 | 0.8633 | 0.8641 | 0.8636 | 0.8784 |
| **d4_dropout** | **0.8349** | **0.8402** | **0.8586** | **0.8450** | **0.8430** | **0.8443** |
| Duplicate | 0.8887 | 0.8740 | 0.8974 | 0.8794 | 0.8903 | 0.8860 |
| Zero | 0.8917 | 0.8585 | 0.8750 | 0.8825 | 0.9610 | 0.8937 |
| d4_dropout $\Delta\%$ vs BL | $-8.2\%$ | $-5.8\%$ | $-0.5\%$ | $-2.2\%$ | $-2.4\%$ | $-3.9\%$ |

Two findings stand out. First, at a 100K-step budget, `d4_dropout` remains substantially better than the matched baseline, with a larger relative MASE gap ($-3.9\%$ vs. $-2.9\%$ at 10K) and lower cross-seed variance. Second, the capacity controls that were near-neutral at 10K now actively hurt: Duplicate degrades by 0.9% and Zero by 1.7%. The baseline shows higher variance across seeds at 100K (seed 0 reaches 0.9092 due to late-training instability), while `d4_dropout` remains stable (range 0.83–0.86).

### E.1. 10K warmup

To verify that the benefit is not an artifact of our 1K-warmup schedule, we also train with the 10K warmup steps across 5 seeds.

*Table 15.* 10K warmup, 100K steps, 5 seeds. Normalized geomean MASE per seed.

|  | **s0** | **s1** | **s2** | **s7** | **s42** | **Mean** |
|---|---|---|---|---|---|---|
| Baseline | 0.884 | 0.884 | 0.875 | 0.912 | 0.901 | 0.891 |
| `d4_dropout` | **0.886** | **0.840** | **0.841** | **0.875** | **0.879** | **0.864** |
| $\Delta$% vs BL | $+0.2\%$ | $-5.0\%$ | $-3.8\%$ | $-4.0\%$ | $-2.4\%$ | $-3.0\%$ |

`d4_dropout` improves by $-3.0\%$ under the official protocol (4 of 5 seeds show clear gains), consistent with $-3.9\%$ under our default 1K-warmup schedule.

## F. FEV-Bench Evaluation

To verify that our findings generalize beyond GIFT-Eval, we evaluate on FEV-Bench (100 tasks), an independent time series forecasting benchmark, and reproduce the polynomial-degree sweep over $d \in \{2, \dots, 7\}$.

*Table 16.* FEV-Bench headline (5 seeds each). MASE values are unnormalized. Wins: per-task mean comparisons out of 100.

| Method | MASE | SQL | Wins/100 |
|---|---|---|---|
| `d4_dropout` (Ours) | **1.254** | **1.024** | **78/100** |
| Baseline | 1.282 | 1.045 | — |
| $\Delta$% | $-2.2\%$ | $-1.9\%$ | $p < 10^{-7}$ |

`d4_dropout` achieves $\Delta = -2.2\%$ geometric-mean MASE on FEV-Bench (5-seed mean $1.254 \pm 0.006$ vs. BL $1.282 \pm 0.007$) and $\Delta = -1.9\%$ SQL ($1.024 \pm 0.005$ vs. $1.045 \pm 0.004$). The improvement wins on 78/100 tasks ($p < 10^{-7}$, paired sign test), consistent with the GIFT-Eval results.

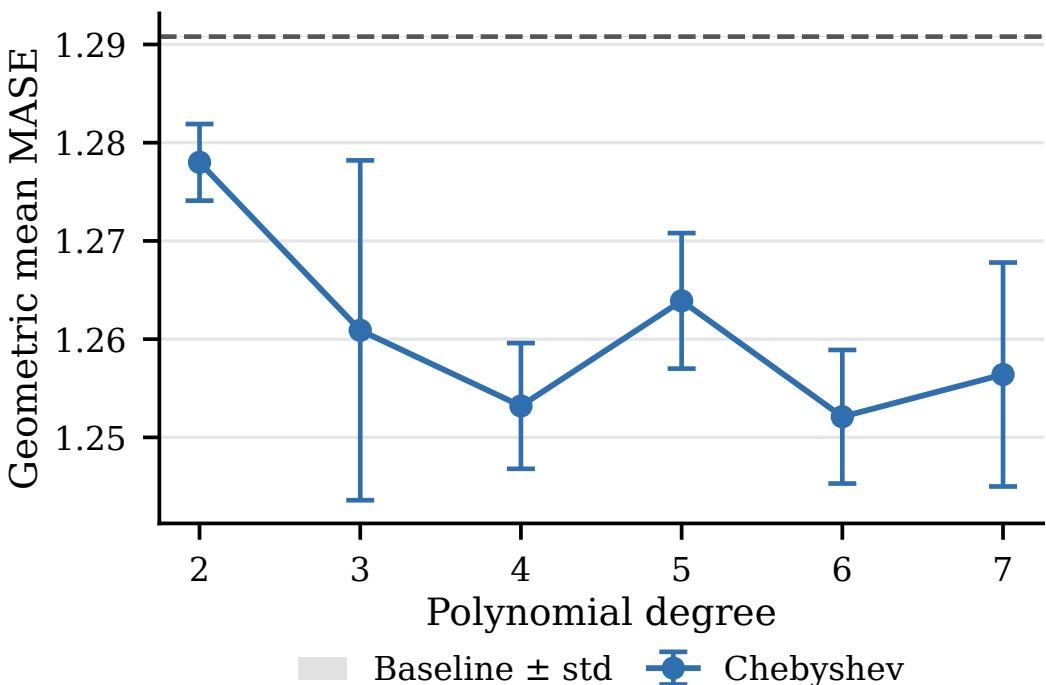

*Figure 4.* FEV-Bench polynomial-degree sweep (5 seeds per $d$, stride $s{=}16$, 10% channel dropout). Errorbars are cross-seed std; dashed line is the 5-seed baseline mean. Every tested $d \geq 2$ beats baseline, mirroring the GIFT-Eval pattern (Fig. 2).

*Table 17.* FEV-Bench polynomial-degree sweep, 5 seeds per degree. $\Delta\%$ relative to the 5-seed baseline mean (1.282).

| Degree | MASE (mean $\pm$ std) | $\Delta\%$ vs BL | SQL (mean $\pm$ std) |
|---|---|---|---|
| 2 | $1.2780 \pm 0.0039$ | $-0.2\%$ | $1.0425 \pm 0.0032$ |
| 3 | $1.2609 \pm 0.0173$ | $-1.6\%$ | $1.0288 \pm 0.0125$ |
| 4 | $1.2532 \pm 0.0064$ | $-2.2\%$ | $1.0236 \pm 0.0053$ |
| 5 | $1.2639 \pm 0.0069$ | $-1.3\%$ | $1.0309 \pm 0.0063$ |
| 6 | $1.2521 \pm 0.0068$ | $-2.3\%$ | $1.0236 \pm 0.0060$ |
| 7 | $1.2564 \pm 0.0114$ | $-1.9\%$ | $1.0263 \pm 0.0094$ |

The FEV-Bench degree-sweep reproduces the qualitative GIFT-Eval pattern: every $d \geq 2$ beats baseline, the curve is shallow, and the extreme $d=2$ (single non-trivial lag) trails the rest. The benefit transfers cleanly to an independent benchmark.

