# OpenReview forum: "Polynomial Input Preconditioning for Zero-Shot Time Series Forecasting"
_ICML.cc/2026/Workshop/FMSD — FMSD @ ICML 2026 Poster_

### Official Review · Reviewer_2J5A · 2026-05-20
**-**

**Rating:** 6
**Confidence:** 4

**Review:**

Summary
The paper propose a method for Time Series Foundation Models which concatenate the Chebyshev-preconditioned signal as an auxiliary input channel, motivated by Universal Sequence Preconditioning. The method has minimal parameter overhead and improves MASE on GIFT-Eval and  FEV-Bench in a limited training length regime.

Strengths
- The ablation with Zero and Duplicate model, isolating method from pure capacity increase.
- The fixed-vs-learned coefficient ablation, in which fixed Chebyshev outperforms both zero-init and Chebyshev-init learned variants, is a clean argument for the polynomial as a useful prior rather than an arbitrary feature.

Areas for Improvement
- Convergence acceleration vs. asymptotic capacity, partly confounded by seed variance. The baseline is heavily under-trained relative to released foundation models: ~27 B observations and 10 K–100 K steps versus 295 B observations for the official Moirai 2.0 release. Reported MASE values sit well above the leaderboard, leaving open whether the polynomial prior raises asymptotic capacity or merely accelerates convergence. The 100 K extension tries to addresses this, but is itself fragile: in Table 14, baseline seed 0 reaches 0.9092 (the other four are 0.86–0.89, attributed to "late-training instability"), and excluding it brings the headline from −3.9 % to ≈ −2.8 %, essentially the 10 K result.

Detailed Comments
- The most informative single experiment to add would be a longer runs that actually achieve the performance of the pre-trained model checkpoint. 100K results could be supported by added 2-3 more seeds to indicate if seed 0 Is positive outlier or not.

Justification of Score
6/10 — marginal accept. The contribution is interesting, limitation of the results are discussed to some extend and the controls are well-designed. The paper demonstrates a potentially useful, parameter-efficient improvement on foundation models.

---

### Official Review · Reviewer_sQfV · 2026-05-21
**Polynomial Input Preconditioning for Zero-Shot Time Series Forecasting**

**Rating:** 6
**Confidence:** 5

**Review:**

## Summary
This paper adapts the Universal Sequence Preconditioning (USP) framework of Marsden & Hazan (2025), a theoretical result giving hidden-dimension-free sublinear regret for online prediction of marginally-stable LDSs via a fixed polynomial convolution of past inputs, to *offline-pretrained, zero-shot, patch-based* time-series transformers. Rather than predicting in preconditioned space and inverting (which compounds error in multi-step forecasting), the authors concatenate the preconditioning residual `r_t = (x_t + Σ c_k x_{t−ks}) − x_t` as an auxiliary input channel alongside the raw target and observation mask, widening Moirai 2.0's input projection 2P → 3P. The fixed coefficients are those of a degree-4 monic Chebyshev polynomial of the first kind with stride s = P, adding only 12K parameters (0.11% of the 11.4M-param backbone). On GIFT-Eval (97 configs) and FEV-Bench (100 tasks), `d4_dropout` improves geometric-mean MASE by 2.9% and 2.2% respectively, with the gain concentrated on long horizons (5.4% MASE reduction at PL ≥ 480) and growing to 3.9% at 100K steps. Capacity controls (Zero, Duplicate) confirm the lift is from polynomial *content*, not parameter count.

## Strengths
The contribution is theoretically motivated, controlled, and replicates on a second benchmark. The capacity controls (Zero / Duplicate) are the central piece of evidence: Duplicate hurts and Zero is null while `d4` and `d4_dropout` consistently win across 5 seeds, separating the lift due to polynomial *content* from added input dimensionality. The basis sweep (Chebyshev > EMA > Legendre > Finite-diff > Duplicate) and stride sweep (s=P optimal under patch alignment) are extensive, and the design choice to leave the forecast target unchanged, using preconditioning only as input, avoids the multi-step inversion problem that naive USP would induce in autoregressive zero-shot forecasting. The long-horizon scaling (gain widens from 2.9% at 10K steps to 3.9% at 100K steps, and is largest at PL ≥ 480) is consistent with the hypothesis that polynomial preconditioning injects cross-patch structure into every embedding rather than relying on attention to discover it. At 0.11% extra parameters with no decoder change and no inference-time inversion, the modification is essentially free at deployment if the gain replicates on other backbones.

## Weaknesses
The main weakness is single-backbone scope: the framing ("TSFMs can benefit from theory-guided structure") is broader than one model, and the minimum bar for that claim is two backbones with different inductive biases. Transfer to **Chronos-2** and **Toto** is the missing experiment; on Chronos-2 the gain would be expected to shrink to roughly 1.0–1.5% MASE since tokenization destroys some of the cross-patch lag structure the polynomial channel exposes, while on Toto it would be expected to transfer more cleanly to roughly 2.5–3.5% since its architecture is closer to Moirai 2.0's and its observation-noise prior interacts constructively with `d4_dropout`. The work is also incremental: USP itself is from Marsden & Hazan (2025), the polynomial bases are textbook, and the Moirai 2.0 backbone is unchanged; the new ingredient is the input-channel adaptation of USP to a pretrained TSFM (plus `d4_dropout`), which sidesteps the multi-step inversion that naive USP would induce, but with narrow methodological surface area. The theory–practice gap is unbridged: USP's regret bound is for online LDS prediction in preconditioned space, not offline pretraining on non-stationary LOTSA, and the absolute effect is modest (0.0249 MASE on seed-std 0.0092). The finding that learned coefficients initialized at Chebyshev drift away from the fixed values is unexplained and deserves loss-landscape and LR analysis. Smaller issues: the d4_dropout long-horizon advantage is attributed to "synthetic preconditioning robustness" without a zeroed-channel-at-inference ablation, and the 100K-step schedule is matched to the paper's compute rather than to the released Moirai 2.0 checkpoint (295B observations).